# A Dynamic Traffic Assignment Model for the Sustainability of Pavement Performance

**Xinhua Mao [1,2,\*], Jianwei Wang [1], Changwei Yuan [1], Wei Yu [2,3] and Jiahua Gan [4]**

[1]   School of Economics and Management, Chang'an University, Xi'an 710064, China; wjianwei@chd.edu.cn (J.W.); changwei@chd.edu.cn (C.Y.)

[2]   Department of Civil and Environmental Engineering, University of Waterloo, Waterloo, ON N2L 3G1, Canada; changda.yuwei@gmail.com

[3]   College of Traffic and Transportation, Chongqing Jiaotong University, Chongqing 400074, China

[4]   Transport Planning and Research Institute, Ministry of Transport, Beijing 100028, China; ganjh@tpri.org.cn

\*   Correspondence: mxinhua@uwaterloo.ca

**Abstract:** Existing Dynamic Traffic Assignment (DTA) models assign traffic flow with the principle of travel time, which are easy to distribute most of the traffic flows on the shortest path. A serious unbalance of traffic flow in the network can speed up pavement deterioration of highways with heavy traffic, which influences the sustainability of pavement performance and increases maintenance expenditures. The purpose of this research is to obtain a more optimized traffic assignment for pavement damage reduction by establishing a multi-objective DTA model with the objectives of not only minimum travel time but minimum decline of Present Serviceability Index (PSI) for pavements. Then, teaching-learning-based optimization (TLBO) algorithm is utilized to solve the proposed model. Results of a case study indicate that a more balanced traffic flow assignment can be realized by the model, which can effectively reduce average PSI loss, save maintenance expenditures, extend pavement service life span, save fuel consumption and reduce pollutant emissions in spite of a little increase of average travel time. Additionally, sensitivity of weight factor for the two objective functions is analyzed. This research provides some insights on methods on sustainable pavement performance.

**Keywords:** sustainability of pavement performance; PSI; maintenance expenditures; pavement service life span; fuel consumption; pollutant emissions; travel time; DTA model; TLBO

## 1. Introduction

As a large infrastructure, pavements are usually constructed with materials that exhibit distresses after construction due to various loading [1]. These distresses include several types like cracking, rutting, potholes, patching, bleeding, surface deterioration [2], which decrease the pavement performance, increase maintenance expenditures, shorten the pavement service life span, and influence driving safety. To keep the pavement performance in a good level, the government cannot but spend a huge amount of capital on the pavement maintenance and rehabilitation every year. Statistics from Ministry of Transport (MOT) in China reveals that the pavement maintenance expenditures for freeways in 2007 were US$7.72 billion with an average of US$47.23 thousand per kilometer, and in the past 5 years, the growth rate of maintenance expenditures was around 6.5% [3]. It can be estimated that with the continuous expansion of network scale and the rapid growth of vehicles, especially trucks, pavement deterioration rate will increase with a higher speed in China. Actually, how to improve the sustainability of pavement performance has become a challenge for Chinese government.

Obvious unbalance of traffic flows can be always observed among different highways towards the same destination in a network, and pavement performance of these highways with heavier traffic deteriorates more quickly than others, which must need a usual maintenance work and may cost more maintenance expenditures. An optimal traffic assignment with a balance between the highways with heavy traffic and those with light traffic is an effective way to relieve pavement performance deterioration and to save maintenance expenditures for the whole network. But, how to identify the vehicle types and traffic volumes on every highway is a key concern. As we know, traffic load is the most important factor which controls the pavement performance as well as the maintenance cost of the road [4]. Each individual axle with a specific weight and configuration of vehicles can be converted into an equivalent number of standard axle (an 18-kip single axle) loads (ESALs) [5], which is used to calculate traffic load. Present Serviceability Index (PSI) is widely used to measure pavement performance [6]. Lots of empirical researches have proved that a typical power function can be used to analyze the correlation between PSI and ESALs [7], which provides a technique to control pavement performance deterioration. Regulating the distribution of ESALs for highways by assigning all types of vehicles in the network is the idea of this research.

Dynamic traffic assignment (DTA) refers to a method assigning trips of given aggregate origin-destination (OD) pair traffic demands to the routes in the network with a specific principle [8–10], which plays an important role in decision-making towards various transportation policies [11]. In the current practice, various methods and models have been developed for DTA, which can be categorized into (i) mathematical programming model (MPM) [12–14], (ii) optimal control model(OCM) [15–17], and (iii) variational inequality model (VIM) [18–20]. MPM is a linear or nonlinear mathematical programming model with a hypothesis that the travel time is discrete. For instance, Ziliaskopoulos used a cell transmission model to formulate DTA problem, which proved that DTA problem can be modeled as a linear program, but the model was limited to a network with single destination [21]. Birge and Ho established a linear stochastic DTA model with random traffic demands and congestion limits on traffic flow, and the case study showed the model can obtain a globally optimal solution by a decomposition algorithm [22]. Janson presented a nonlinear DTA mode of user-equilibrium for urban road networks with complex traffic demands, which can generate both static and dynamic assignments that approximately satisfy the user-equilibrium conditions [23]. OCM assumes that the travel time of vehicles varies continuously with traffic volume of a highway segment, namely, travel time can be represented by a continuous function of traffic volume. For instance, Wie et al. used an equivalent continuous time optimal DTA model with steady-state assumptions to assign traffic flow on a congested multiple origin-destination network [24]. Chow established a system optimal DTA model with departure time choice, setting the minimum total system travel cost as model's objective [25]. As for VIM, it was applied to solve DTA problem by Dafermos firstly [26]. VIM simplifies the mathematical formulations of DTA model and relaxes constraints of DTA model to some unilateral constraints, which can make the model more practical and easier to solve. To get optimal departure time and route choice, Friesz et al. put forward a VIM with continuous travel time equilibrium, using efficiency function of highways and penalty function for early and late arrival [27]. Ran and Boyce formulated a link-based VIM to determine vehicle flows on each highway link at each instant of time, which took the influence of queuing delay on actual travel time into account [28]. Additionally, some Traffic simulators also have been developed to solve DTA problems [29–33]. Available literature does not take pavement performance as a variable in the formulation of DTA, which is easy to distribute most of the traffic flow on the shortest path causing unbalance of traffic flow in the network. This research aims to obtain a more optimized traffic assignment for reducing pavement damage by establishing a multi-objective DTA model taking both pavement performance deterioration and travel time into account.

The remainder of this paper is organized as follows. Section 2 introduces a methodology in detail used in this research. Section 3 describes a case study including the data collection and result analysis.

Section 4 discusses the sensitivity of weight factor for the two objective functions. The contribution and future work of the proposed research are presented in Section 5.

## 2. Methodology

### 2.1. General

As mentioned in the Introduction, the problem of traffic assignment in network can be solved by DTA models. Existing DTA models assign traffic with the principle of minimum travel time. A basic DTA model is formulated as in [34].

$$\min Z = \sum_a \int_0^{v_a} s_a(v)dv \tag{1}$$

Subject to the constraints:

$$\sum_k h_{k,pq} = g_{pq}, \text{ all } p \in \{O\}, \ q \in \{D\} \tag{2}$$

$$h_{k,pq} \geq 0, \text{ all } k \in K_{pq}, \ p \in \{O\}, \ q \in \{D\} \tag{3}$$

$$v_a = \sum_{pqk} h_{k,pq} \cdot \delta_{ak,pq} \text{ all } a \in \{A\} \tag{4}$$

where, $Z$ is the objective function; $\{O\}$, $\{D\}$ and $\{A\}$ are the sets of origin nodes, destination nodes and direct links respectively; $p$, $q$ and $a$ are arbitrary elements of $\{O\}$, $\{D\}$ and $\{A\}$ respectively; $v_a$ is the traffic flow on link $a$; $s_a(v)$ is a traffic delay function for link $a$; $h_{k,pq}$ is trips per unit time on $k$ from $p$ to $q$; $K_{pq}$ denotes a set of paths from $p$ to $q$; $k$ is an arbitrary element of $K_{pq}$; $g_{pq}$ is the traffic demand per unit time from $p$ to $q$; $\delta_{ak,pq}$ is 1, if link $a$ lies on path $k$ from $p$ to $q$ and 0, otherwise.

To some extent, traffic assignment with the principle of minimum travel time conforms to the reality of vehicle routing choice without traffic control, but can bring about an obvious unbalance of traffic flows on several highways towards the same direction. A simple experiment is carried out in a network with a single OD and three routes, shown in Figure 1, using the above DTA model in this research. Assume that traffic demands from O to D are 2000 vehicles, the lengths of $R_1$, $R_2$ and $R_3$ are $L_1 = 3.6$ km, $L_2 = 2.8$ km and $L_3 = 3.8$ km. Assignment results indicate that traffic flows on $R_1$, $R_2$ and $R_3$ are $v_1 = 500$ vehicles, $v_2 = 1100$ vehicles, and $v_3 = 400$ vehicles, which means that 55% of vehicles are assigned to the shortest route $R_2$ causing an unbalanced traffic distribution in the network. Correspondingly, pavement deterioration rate of highways on route $R_2$ may be higher than that on the other two routes $R_1$ and $R_3$. To solve the problem effectively, the DTA model should be modified with the consideration of pavement deterioration to obtain a more optimized traffic assignment.

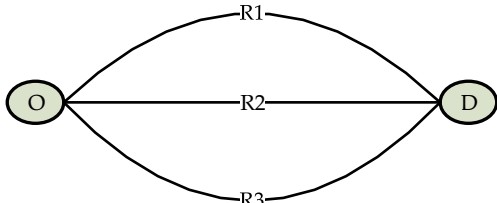

**Figure 1.** A network with a single OD and three routes.

### 2.2. Pavement Deterioration Model

Pavement performance can be defined in different ways. The concept of PSI is widely used in evaluating the pavement performance and forecasting the pavement failure [35], which is defined as the ability of a specific section of pavement to serve high speed, high volume, and mixed traffic in its existing condition [36]. Cumulative ESALs can be used to measure the traffic load caused by vehicles

to pavements [37]. For different types of pavements, the correlation between PSI and cumulative ESALs can be identified using Equation (5).

$$\text{PSI} = \frac{P_0}{P_0 - P_c} \cdot \alpha \cdot ESALs^\beta \tag{5}$$

where $P_0$ is initial PSI; $P_c$ is terminal PSI of 2.5; $\alpha$ and $\beta$ are field calibration coefficients for a specific pavement type; *ESALs* is the cumulative equivalent single axle loads during the analysis period, which can be calculated as follows.

$$ESALs = \sum_{i=1}^{n} \sum_{j=1}^{m} Q_{ij} \cdot LEF_{ij} \tag{6}$$

where $Q_{ij}$ is the traffic volume of vehicle class $i$ ($I = 1 - n$), which belongs to axle weight group $j$ ($j = 1 - m$); $LEF_{ij}$ is load equivalency factor of weight group $j$ in vehicle class $i$.

### 2.3. Multi-objective DTA Model with Pavement Performance Consideration

Since PSI is considered in this research as an important variable to control traffic volumes of all routes within reasonable limits, which makes sure that it will not accelerate the pavement performance deterioration. While the principle of travel time helps assign traffic flow on optimal routes. So, a multi-objective DTA model is established as follows.

$$\text{Minimize } T = \sum_{s=1}^{N} \sum_{i=1}^{n} \sum_{j=1}^{m} \int_{0}^{Q_{sij}} f_s(Q_{ij}) d_{Q_{ij}} \tag{7}$$

$$\text{Minimize } P = \sum_{s=1}^{N} [P_s^0 - \frac{P_s^0}{P_s^0 - P_s^c} \cdot \alpha_s \cdot (\sum_{i=1}^{n} \sum_{j=1}^{m} Q_{sij} \cdot LEF_{sij})^{\beta_s}] / N \tag{8}$$

Subject to the constraints:

$$\sum_{k=1}^{K_w} \sum_{i=1}^{n} \sum_{j=1}^{m} h_k^{wij} = q^w, \forall w \in W \tag{9}$$

$$\sum_{k=1}^{K_w} \sum_{i=1}^{n} \sum_{j=1}^{m} h_k^{wij} \cdot \delta_s^{wkij} = Q_s, \forall s \in N, \forall w \in W \tag{10}$$

$$\frac{P_s^0}{P_s^0 - P_s^c} \cdot \alpha_s (\sum_{i=1}^{n} \sum_{j=1}^{m} Q_{sij} \cdot LEF_{sij})^{\beta_s} \geq \tau_s, \forall s \in N \tag{11}$$

$$h_k^{wij} \geq 0, \forall k \in K_w \tag{12}$$

where, $T$ is total travel time; $s$ is highway segment $s$, $s = 1, 2, \cdots, N$; $i$ is vehicle class $i$, $i = 1, 2, \cdots, n$; $j$ is axle weight group $j$, $j = 1, 2, \cdots, m$; $Q_{sij}$ is traffic flow of vehicle class $i$ in axle weight group $j$ on highway segment $s$; $f_s(Q_{ij})$ is delay function for highway segment $s$ related to traffic flow $Q_{ij}$, which can be formulated by the function proposed by the Bureau of Public Roads (BPR), whose role is currently performed by the Federal Highway Administration (FHWA) [38]; $P$ is the average PSI decline of all highway segments at the end of research period; $P_s^0$ is initial PSI of highway segment $s$; $P_s^c$ is terminal PSI of highway segment $s$; $\alpha_s$ and $\beta_s$ are field calibration coefficients of highway segment $s$ according to its specific pavement type; $LEF_{sij}$ is load equivalency factor of vehicle class $i$ in axle weight group $j$ on highway segment $s$; $W$ is a set of OD pairs; $w$ is OD pair $w$, $w = 1, 2, \cdots, W$; $K_w$ is a set of paths of OD pair $w$; $k$ is path $k$, $k = 1, 2, \cdots, K_w$; $h_k^{wij}$ is traffic flow of vehicle class $i$ in axle weight group $j$ on path $k$ of OD pair $w$; $q^w$ is traffic need of OD pair $w$; $\delta_s^{wkij}$ is 1, if highway segment $s$ with traffic flow of vehicle class $i$ in axle weight group $j$ lies on path $k$ of OD pair $w$ and 0, otherwise; $\tau_s$ is lower limit value of PSI of highway segments at the end of the research period.

As Equation (7), one objective function of the model helps assign trips to network links such that the network-wide total travel time is minimized. As another objective function of the model, Equation (8) makes sure that the average PSI decline of all highway segments will keep at a lowest level at the end of the research period to reduce maintenance and rehabilitation expenditures. Equation (9) is

the traffic flow conservation constraint that indicates the traffic need of a specific OD pair is the sum of traffic flow on all paths linking the OD pair; Equation (10) indicates that the traffic volume on a specific highway segment is the sum of traffic volume on all paths on which the highway segment lies; Equation (11) sets the lower limit value of PSI for every highway segment an the end of research period; Equation (12) sets lower bound of assigned traffic coming from a specific O-D pair.

*2.4. Model Solution*

The proposed multi-objective DTA model with pavement performance consideration is a continuous non-linear large scale problem [39], which can be solved by several traditional heuristic algorithms like Genetic Algorithm (GA) [40,41], Artificial Immune Algorithm (AIA) [42], Ant Colony Optimization (ACO) [43], Particle Swarm Optimization (PSO) [44], Gravitational Search Algorithm (GSA) [45], etc. But, the above algorithms have the limitation that different parameters must be set during the computational process.

Rao et al. firstly proposed TLBO algorithm [46], which has the advantage of requiring only common controlling parameters for its working and fast convergence [47]. TLBO simulates the teaching-learning process of teachers and students in class rooms, which is divided into two phases, known as teacher phase and learner phase [48]. Teachers provide different subjects to a group of learners and train the learners to achieve better results in terms of grades. Moreover, learners learn from each other to improve their grades. In TLBO, a group of learners is considered as population, different subjects are considered as variables, and the learner with highest grades is considered as the best solution [49]. As for the proposed multi-objective DTA model, TLBO can be defined mathematically as follows.

Normalize the multi-objective function $Z$ considering different weight factors to both the objective functions.

$$Z = \text{Minimize } F(X) = \theta\left(\frac{T}{T_{min}}\right) + (1 - \theta)\left(\frac{P}{P_{min}}\right) \tag{13}$$

where, $\theta$ is a weight factor for the first objective function $T$, $\theta \in [0,1]$; $P_{min}$ and $T_{min}$ are the minimum values of objective functions $P$ and $T$ respectively when these objectives are considered independently; $F(X)$ is the normalized objective function, and $X$ is a multi-dimensional vector for variables.

Assume that $X^{\gamma} = (x_1^{\gamma}, x_2^{\gamma}, \cdots, x_d^{\gamma})$, $\gamma = 1, 2, \cdots, V$, $X_{\phi} = (x_{\phi}^1, x_{\phi}^2, \cdots, x_{\phi}^V)$, $\phi = 1, 2, \cdots, D$ and $x_{\phi}^{\gamma}$ is learner $\gamma$ in subject $\phi$. Where, $D$ is the number of subjects; $V$ is the population of learners. Then, the class room can be formulated as.

$$\begin{bmatrix} X^1 & F(x^1) \\ X^2 & F(x^2) \\ \vdots & \vdots \\ X^V & F(x^V) \end{bmatrix} = \begin{bmatrix} x_1^1 & x_2^1 & \cdots & x_D^1 & F(X^1) \\ x_1^2 & x_2^2 & \cdots & x_D^2 & F(X^2) \\ \vdots & \vdots & \cdots & \vdots & \vdots \\ x_1^V & x_2^V & \cdots & x_D^V & F(X^V) \end{bmatrix} \tag{14}$$

The iterative steps of the TLBO are as following eight steps.

Step1: Initialize the population of learners, $\gamma = 1, 2, \cdots, V$ and numbers of subjects offered to the learners, $\phi = 1, 2, \cdots, D$ with random generation and evaluate them.

Step2: Designate the best learner $F(X)_{\phi,best}$ as chief teacher for that iteration in subject $\phi$. $X_{\phi,teacher} = F(X)_{\phi,best}$.

Step3: Calculate current mean values $M_{\phi}$ of learners in each subject $\phi$. The teacher in each subject $\phi$ acts as a new mean and tries to shift the mean from $M_{\phi}$ to $M\_new_{\phi}$, $M\_new_{\phi}$, $= X_{\phi,teacher}$.

$$M_{\phi} = \frac{1}{V} \sum_{\gamma=1}^{V} x_{\phi}^{\gamma} \tag{15}$$

Step4: Calculate the difference between $M_{\phi}$ and $M\_new_{\phi}$ using the teaching factor $TF_{\phi}$.

$$Difference_{\phi} = r_{\phi}(X_{\phi,teacher} - TF_{\phi} \cdot M_{\phi}) \tag{16}$$

$$TF_{\phi} = round[1 + rand(0,1)], \; r_{\phi} = rand(0,1) \tag{17}$$

Step5: Update each learner's knowledge $x^{\gamma}_{\phi,old}$ with the help of teacher's knowledge according to

$$x^{\gamma}_{\phi,new} = x^{\gamma}_{\phi,old} + difference_{\phi} \tag{18}$$

Step6: Update the learners' knowledge using the knowledge acquired by the learners during the tutorial hours, according to

$$x^{\gamma}_{\phi,new} = x^{\gamma}_{\phi,old} + r_{\phi}(x^{\gamma}_{\phi,} - x^{\epsilon}_{\phi}), \text{ if } F(X^{\gamma}) < F(X^{\epsilon}), \; \gamma, \epsilon = 1, 2, \cdots, V, \; \gamma \neq \epsilon \tag{19}$$

$$x^{\gamma}_{\phi,new} = x^{\gamma}_{\phi,old} + r_{\phi}(x^{\epsilon}_{\phi,} - x^{\gamma}_{\phi}), \text{ if } F(X^{\epsilon}) < F(X^{\gamma}), \gamma, \epsilon = 1, 2, \cdots, V, \; \gamma \neq \epsilon \tag{20}$$

Step7: Combine all the subjects.

Step 8: Repeat the procedure from steps 2 to 7 till the termination criterion is met.

## 3. Case Study

A case study using the arterial network in Chengdu Metropolitan Area (CMA), including Chengdu, Ziyang, Meishan, and Leshan, four neighboring cities in Sichuan Province in China, as testbed was considered. Figure 2 shows the highway network consisting of 4 National Highways marked by G and numbers and 4 Provincial Highways marked by S and numbers. National Highways have three 3.75 m lanes per direction with a speed limit of 120 km/h and designed traffic capacity of 2200 pcu/h/ln. While Provincial Highways have two 3.75 m lanes per direction with a speed limit of 100 km/h and designed traffic capacity of 2000 pcu/h/ln. The 8 highways are all designed as flexible pavements using asphalt mixture.

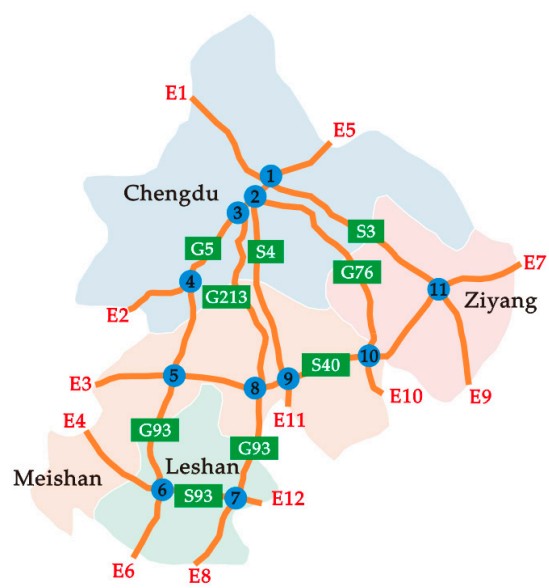

**Figure 2.** Arterial highway network geometry in CMA.

G5, G93 and G76 are the three most important highways with the maximum traffic volume in the network, which carry both transit and internal traffic. Available traffic data provided by Sichuan Highway Administration Bureau (SHAB) indicates that annual average daily traffic (AADT) for the three highways are 36,021 vehicles per day, 34,567 vehicles per day and 34,215 vehicles per day on both directions respectively, accounting for about 60% traffic volume of the whole highway network.

Additionally, the traffic stream of the three highways carries around 22% heavy vehicles throughout the day, causing an increasing decline of pavement performance. So, to keep the highways at a good service level, SHAB spends US$46.1 million on the maintenance for the whole network every year, which brought about a heavy financial burden.

The network was described as a topology graph including 11 nodes and 27 arcs (H1 to H27), shown in Figure 3, where nodes E1 to E12 refer to the external origins and destinations of vehicles, and nodes 1 to 11 indicate the 11 intersections of highways in the network.

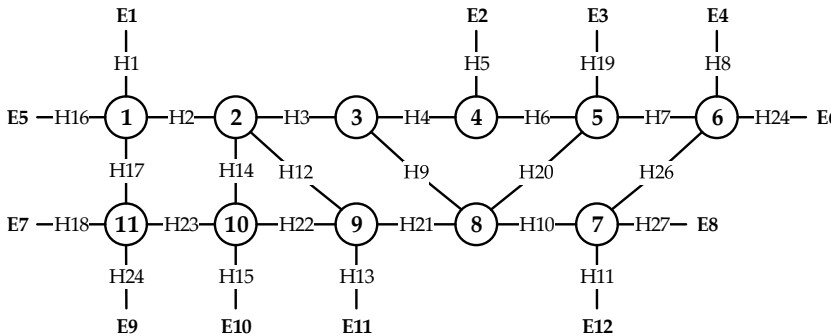

**Figure 3.** Topology graph of the arterial highway network in CMA.

### 3.1. Data Collection and Processing

Data of highway network design, geometrics, travel demand, traffic operations, and initial pavement performance is needed for this computational study.

### 3.1.1. Basic Data

Length of highways, number of lanes, traffic capacity and pavement type were collected from the Highway Geographic Information System managed by SHAB.

Traffic volume, vehicle class and axle loads were collected at the 11 nodes respectively by automatic traffic recorders (ATRs), automatic vehicle classifiers (AVCs), and weigh-in-motion (WIM) scales. In this study, vehicles were classified into 13 classes according to FHWA classification scheme, displayed in Table 1. And axle loads for different vehicle classes were classified into 10 groups [50]: 2 kips single, 6 kips single, 12 kips single, 18 kips single, 22.4 kips single, 30 kips single, 24 kips tandem, 32 kips tandem, 40 kips tandem and 48 kips tandem.

**Table 1.** FHWA Vehicle Classes for Heavy Vehicles.

| Vehicle Class | Description |
| --- | --- |
| Class 1 | Motorcycles |
| Class 2 | Passenger cars |
| Class 3 | Two-axle, four-tire single units |
| Class 4 | Buses |
| Class 5 | Two-axle, six-tire single units |
| Class 6 | Three-axle single units |
| Class 7 | Four or more axle single units |
| Class 8 | Four or less axle single trailers |
| Class 9 | Five-axle single trailers |
| Class 10 | Six or more axle single trailers |
| Class 11 | Five or less axle multitrailers |
| Class 12 | Six-axle multitrailers |
| Class 13 | Seven or more axle multitrailers |

O-D data of different vehicle classes and axle loads was collected by Vehicle License Plate Recognition (VLPR). License information of every vehicle was collected twice respectively when it ran

into and out of the highway network on its passing nodes, so that which node the vehicle was from and which node it was to can be identified.

Typical flexible pavement distresses were observed during the road test implemented by SHAB, and initial PSI was calculated by the testing data of these typical flexible pavement distresses. Calibrated equation used in this study is as follow.

$$PSI = 5.11 - 1.94 \cdot \log\left(1 + \overline{SV}\right) - 1.29\overline{RD}^2 - 0.03\sqrt{C + G} \tag{21}$$

where, $\overline{SV}$ is the mean value of slop variance in the wheel paths; $\overline{RD}$ is the mean rut depth (in); $C$ is cracking (ft$^2$/1000ft$^2$); $G$ is patching (ft$^2$/1000ft$^2$).

### 3.1.2. Parameter Calibration

The coefficients $\alpha$ and $\beta$ in Equation (5) have different values for different pavement types. In this case study, a single type of flexible pavement is used. Therefore, $\alpha$ and $\beta$ are constant values that do not differ between the highway segments considered, and are calibrated using a natural logarithm as follows.

$$\log PSI = \log\left[\frac{P_0}{P_0 - P_c} \cdot \alpha \cdot ESALs^{\beta}\right] \tag{22}$$

$$\log \frac{PSI \cdot (P_0 - P_c)}{P_0} = \log \alpha + \beta \cdot \log ESALs \tag{23}$$

Denoting that $Y = \log\frac{PSI \cdot (P_0 - P_c)}{P_0}$, and $X = \log ESALs$, Equation (23) can be transformed to a linear function between $Y$ and $X$.

$$Y = \beta \cdot X + \log \alpha \tag{24}$$

To calibrate the coefficients $\alpha$ and $\beta$, field data PSI, $P_0$ and $ESALs$ of the highway network were collected. For a highway segment, PSI is the PSI of the highway segment when the $\xi^{\text{th}}$ maintenance is implemented. $P_0$ is the PSI of the highway segment after the $(\xi - 1)^{\text{th}}$ maintenance. $ESALs$ is the cumulative ESALs during the period between the $(\xi - 1)^{\text{th}}$ and $\xi^{\text{th}}$ maintenance. And all these data needed was extracted from Highway Maintenance Database established by SHAB. We extracted 53 maintenance histories of the highway network in the past 7 years.

Figure 4 presents the calibrated function using field data of 53 maintenance histories. Values of $\alpha$ and $\beta$ in this research are $-4.125 \times 10^{-12}$ and $-0.865$.

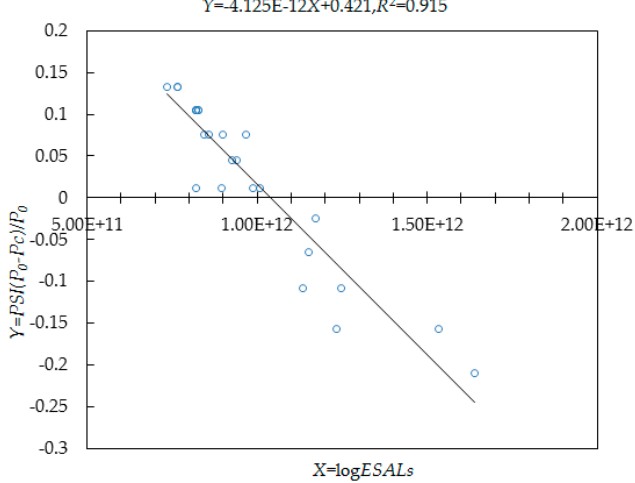

**Figure 4.** Calibrated function for PSI of flexible pavement.

*3.2. Results*

The proposed model and algorithm are coded in the Microsoft Windows-based MATLAB R2018b (version 11.4). In the process of executing the proposed algorithm, initial inputs, parameters, and number of iterations are set as $V = 20$, $D = 4$, $\theta = 0.5$, number of iterations is 30, $r_\phi = 0.6$, $TF_\phi = 2$.

### 3.2.1. Distribution of Traffic Flow

The outcome of the model is a new distribution of different types of vehicles in the network. It can be obtained from the comparison between model results and actual traffic distribution that some of the vehicles have a route diversion. The distribution of three light vehicle classes (Classes 1, 2, and 3) remains unchanged, because they have light axle loads and their damage to flexible pavement can be negligible [51]. Vehicles of class 4 have no route diversion, because they have a small degree of damage and they mainly provide commuting services for residents, which has a high demand of short travel time. So, these vehicles still travel along the actual most time-saving routes. Vehicles of class 12 and 13 have no route diversion too, because they account for a very small proportion about 0.5% of AADT and it is safer for them to travel on the current highways with 3 lanes per direction. But, vehicles of class 5 to 11 have a significant route diversion, which implies that they are the most important factors causing pavement damage.

Tables 2 and 3 summarize the highway segments involved with traffic flow changes on upstream direction (from south to north and from east to west) and on downstream direction (from north to south and from west to east) respectively. As shown in Tables 2 and 3, traffic flows of class5 to 11 change on highway segments except H1, H16 and H24. There is a decline of traffic flows of class 5 to 11 on G5, G93 and G76, but an increase on the other highways, namely, some vehicles divert their routes from G5, G93 and G76 to the adjoining highways. Of all diverting vehicles, vehicles of class 5, 8, 9, 10 account for about 75% on both directions. From the comparison between route diversions of vehicles on the two directions, we can also know that more vehicles have route diversions on downstream direction than that on upstream direction, which matches the traffic characteristics on arterial highways in China. Because mineral resources are mainly in the northwestern China, there is more truck transportation on the downstream direction.

**Table 2.** Highway segments involved with traffic flow changes on upstream direction (vehicles per day).

| Highway | Segment | Class 5 | Class 6 | Class 7 | Class 8 | Class 9 | Class 10 | Class 11 | Total |
|---|---|---|---|---|---|---|---|---|---|
| | H1 | 0 | 0 | 0 | 0 | 0 | 0 | 0 | 0 |
| | H2 | −145(19.28%) | −99(13.16%) | −76(10.11%) | −141(18.75%) | −152(20.21%) | −124(16.49%) | −15(1.99%) | −752(100%) |
| G5 | H3 | −131(19.21%) | −93(13.64%) | −59(8.65%) | −124(18.18%) | −145(21.26%) | −118(17.3%) | −12(1.76%) | −682(100%) |
| | H4 | −146(21.35%) | −96(14.04%) | −65(9.5%) | −127(18.57%) | −118(17.25%) | −117(17.11%) | −15(2.19%) | −684(100%) |
| | H5 | −15(8.93%) | −25(14.88%) | −26(15.48%) | −42(25%) | −23(13.69%) | −31(18.45%) | −6(3.57%) | −168(100%) |
| | H6 | −134(19.03%) | −76(10.8%) | −79(11.22%) | −128(18.18%) | −142(20.17%) | −134(19.03%) | −11(1.56%) | −704(100%) |
| G93 | H7 | −129(18.25%) | −72(10.18%) | −68(9.62%) | −142(20.08%) | −163(23.06%) | −121(17.11%) | −12(1.7%) | −707(100%) |
| | H8 | −131(18.74%) | −102(14.59%) | −31(4.43%) | −134(19.17%) | −158(22.6%) | −127(18.17%) | −16(2.29%) | −699(100%) |
| | H9 | 41(15.07%) | 35(12.87%) | 34(12.5%) | 56(20.59%) | 58(21.32%) | 36(13.24%) | 12(4.41%) | 272(100%) |
| G213 | H10 | 153(16.85%) | 116(12.78%) | 109(12%) | 181(19.93%) | 186(20.48%) | 128(14.1%) | 35(3.85%) | 908(100%) |
| | H11 | 108(18.52%) | 79(13.55%) | 46(7.89%) | 114(19.55%) | 135(23.16%) | 86(14.75%) | 15(2.57%) | 583(100%) |
| S4 | H12 | 96(17.81%) | 56(10.39%) | 58(10.76%) | 102(18.92%) | 116(21.52%) | 98(18.18%) | 13(2.41%) | 539(100%) |
| | H13 | 147(21.09%) | 98(14.06%) | 94(13.49%) | 119(17.07%) | 109(15.64%) | 112(16.07%) | 18(2.58%) | 697(100%) |
| G76 | H14 | −124(24.7%) | −112(22.31%) | −73(14.54%) | 108(−21.51%) | −134(26.69%) | −156(31.08%) | −11(2.19%) | −502(100%) |
| | H15 | −116(18.68%) | −67(10.79%) | −68(10.95%) | −124(19.97%) | −121(19.48%) | −112(18.04%) | −13(2.09%) | −621(100%) |
| | H16 | 0 | 0 | 0 | 0 | 0 | 0 | 0 | 0 |
| S3 | H17 | 127(27.31%) | 106(22.8%) | 74(15.91%) | 135(29.03%) | 161(34.62%) | −156(−33.55%) | 18(3.87%) | 465(100%) |
| | H18 | 0 | 0 | 0 | 0 | 0 | 0 | 0 | 0 |
| | H19 | 13(11.82%) | 12(10.91%) | 13(11.82%) | 18(16.36%) | 24(21.82%) | 19(17.27%) | 11(10%) | 110(100%) |
| | H20 | 18(15.93%) | 21(18.58%) | 14(12.39%) | 16(14.16%) | 15(13.27%) | 21(18.58%) | 8(7.08%) | 113(100%) |
| S40 | H21 | 127(16.58%) | 116(15.14%) | 74(9.66%) | 127(16.58%) | 148(19.32%) | 164(21.41%) | 10(1.31%) | 766(100%) |
| | H22 | 26(9.59%) | 47(17.34%) | 32(11.81%) | 41(15.13%) | 56(20.66%) | 61(22.51%) | 8(2.95%) | 271(100%) |
| | H23 | 43(12.61%) | 63(18.48%) | 61(17.89%) | 56(16.42%) | 58(17.01%) | 52(15.25%) | 8(2.35%) | 341(100%) |
| | H24 | 0 | 0 | 0 | 0 | 0 | 0 | 0 | 0 |
| | H25 | 12(24.49%) | 5(10.2%) | 5(10.2%) | 7(14.29%) | 9(18.37%) | 8(16.33%) | 3(6.12%) | 49(100%) |
| S93 | H26 | 9(11.25%) | 12(15%) | 13(16.25%) | 11(13.75%) | 15(18.75%) | 14(17.5%) | 6(7.5%) | 80(100%) |
| | H27 | 5(12.5%) | 4(10%) | 7(17.5%) | 6(15%) | 6(15%) | 8(20%) | 4(10%) | 40(100%) |

Note: Positive numbers refer to increased vehicles, and negative numbers refer to reduced vehicles.

**Table 3.** Highway segments involved with traffic flow change on downstream direction (vehicles per day).

| Highway | Segment | Class 5 | Class 6 | Class 7 | Class 8 | Class 9 | Class 10 | Class 11 | Total |
|---|---|---|---|---|---|---|---|---|---|
| G5 | H1 | 0 | 0 | 0 | 0 | 0 | 0 | 0 | 0 |
| | H2 | −167(19.56%) | −112(13.11%) | −86(10.07%) | −156(18.27%) | −180(21.08%) | −134(15.69%) | −19(2.22%) | −854(100%) |
| | H3 | −150(19.53%) | −101(13.15%) | −77(10.03%) | −140(18.23%) | −162(21.09%) | −121(15.76%) | −17(2.21%) | −768(100%) |
| | H4 | −187(23.17%) | −96(11.9%) | −82(10.16%) | −150(18.59%) | −149(18.46%) | −128(15.86%) | −15(1.86%) | −807(100%) |
| | H5 | −21(11.17%) | −22(11.7%) | −25(13.3%) | −55(29.26%) | −25(13.3%) | −31(16.49%) | −9(4.79%) | −188(100%) |
| G93 | H6 | −144(17.43%) | −97(11.74%) | −86(10.41%) | −161(19.49%) | −174(21.07%) | −150(18.16%) | −14(1.69%) | −826(100%) |
| | H7 | −155(18.56%) | −94(11.26%) | −91(10.9%) | −164(19.64%) | −179(21.44%) | −134(16.05%) | −18(2.16%) | −835(100%) |
| | H8 | −147(18.89%) | −112(14.4%) | −27(3.47%) | −150(19.28%) | −188(24.16%) | −140(17.99%) | −14(1.8%) | −778(100%) |
| G213 | H9 | 50(16.18%) | 38(12.3%) | 44(14.24%) | 64(20.71%) | 59(19.09%) | 46(14.89%) | 8(2.59%) | 309(100%) |
| | H10 | 198(17.38%) | 156(13.7%) | 131(11.5%) | 220(19.32%) | 232(20.37%) | 163(14.31%) | 39(3.42%) | 1139(100%) |
| | H11 | 131(20.37%) | 90(14%) | 48(7.47%) | 126(19.6%) | 136(21.15%) | 98(15.24%) | 14(2.18%) | 643(100%) |
| S4 | H12 | 104(17.51%) | 69(11.62%) | 62(10.44%) | 116(19.53%) | 125(21.04%) | 108(18.18%) | 10(1.68%) | 594(100%) |
| | H13 | 164(20.79%) | 109(13.81%) | 104(13.18%) | 151(19.14%) | 123(15.59%) | 121(15.34%) | 17(2.15%) | 789(100%) |
| G76 | H14 | −129(17%) | −119(15.68%) | −73(9.62%) | −134(17.65%) | −154(20.29%) | −135(17.79%) | −15(1.98%) | −759(100%) |
| | H15 | −130(18.57%) | −89(12.71%) | −72(10.29%) | −139(19.86%) | −136(19.43%) | −122(17.43%) | −12(1.71%) | −700(100%) |
| S3 | H16 | 0 | 0 | 0 | 0 | 0 | 0 | 0 | 0 |
| | H17 | 133(14.96%) | 135(15.19%) | 93(10.46%) | 150(16.87%) | 181(20.36%) | 174(19.57%) | 23(2.59%) | 889(100%) |
| | H18 | 132(15.53%) | 109(12.82%) | 181(21.29%) | 130(15.29%) | 156(18.35%) | 130(15.29%) | 12(1.41%) | 850(100%) |
| S40 | H19 | 13(10.08%) | 17(13.18%) | 18(13.95%) | 19(14.73%) | 31(24.03%) | 20(15.5%) | 11(8.53%) | 129(100%) |
| | H20 | 15(13.04%) | 22(19.13%) | 13(11.3%) | 16(13.91%) | 15(13.04%) | 22(19.13%) | 12(10.43%) | 115(100%) |
| | H21 | 163(18.27%) | 128(14.35%) | 77(8.63%) | 140(15.7%) | 186(20.85%) | 187(20.96%) | 11(1.23%) | 892(100%) |
| | H22 | 25(7.96%) | 53(16.88%) | 46(14.65%) | 50(15.92%) | 61(19.43%) | 72(22.93%) | 7(2.23%) | 314(100%) |
| | H23 | 47(12.57%) | 81(21.66%) | 60(16.04%) | 61(16.31%) | 62(16.58%) | 56(14.97%) | 7(1.87%) | 374(100%) |
| | H24 | 0 | 0 | 0 | 0 | 0 | 0 | 0 | 0 |
| S93 | H25 | 11(20.75%) | 7(13.21%) | 6(11.32%) | 8(15.09%) | 11(20.75%) | 8(15.09%) | 2(3.77%) | 53(100%) |
| | H26 | 13(13.83%) | 13(13.83%) | 13(13.83%) | 15(15.96%) | 21(22.34%) | 13(13.83%) | 6(6.38%) | 94(100%) |
| | H27 | 5(13.51%) | 4(10.81%) | 8(21.62%) | 7(18.92%) | 6(16.22%) | 5(13.51%) | 2(5.41%) | 37(100%) |

Note: Positive numbers refer to increased vehicles, and negative numbers refer to reduced vehicles.

### 3.2.2. Maintenance Expenditures

Equation (5) shows that PSI can be represented by a negative power function by cumulative ESALs of a flexible pavement, namely, the more cumulative ESALs the pavement has, the higher declining rate its PSI has. A comparison between the different PSI at the end of research period with and without route diversion, shown in Figure 5, indicates that H2, H3, H4, H6, H7, H8, H14, H15 have a significant increase of PSI with reduced vehicles, while other segments with increased vehicles have a little PSI loss.

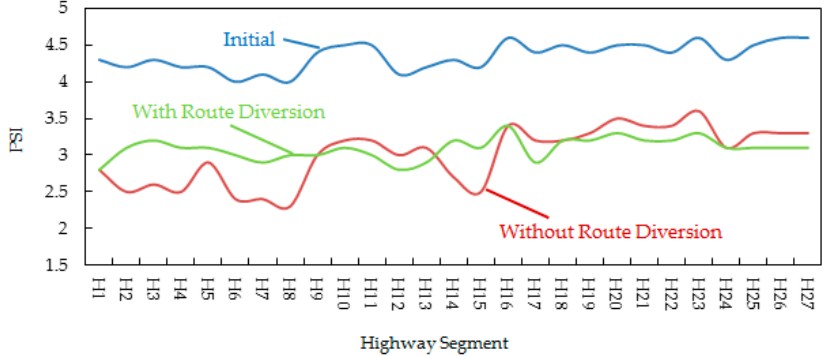

**Figure 5.** PSI at the end of research period.

To keep the pavements in the initial PSI level, maintenance expenditures are needed and can be calculated by the following equation.

$$Expenditure = \frac{\partial Expenditure}{\partial \mathrm{PSI}} \cdot (\mathrm{PSI}_0 - \mathrm{PSI}_c) \cdot L \cdot \sigma \tag{25}$$

where, *Expenditure* is the total network maintenance expenditures needed to keep the pavements in the initial PSI level; $\frac{\partial Expenditure}{\partial \mathrm{PSI}}$ is marginal expenditures caused by unit PSI loss; $\mathrm{PSI}_0$ is the initial PSI; $\mathrm{PSI}_c$ is the terminal PSI; $L$ is the length of the segment; $\sigma$ is the number of lanes of the highway segment.

According to field data of maintenance expenditures for the network, $\frac{\partial Expenditure}{\partial \mathrm{PSI}}$ is calibrated as US$795, when $\mathrm{PSI}_c$ belongs to [2.5, 3) and US$665, when $\mathrm{PSI}_c$ belongs to [3, 3.5). So, the maintenance expenditures with route diversion are US$ 47,698,684, saving US$8,228,161, around 14.71%.

### 3.2.3. Pavement Service Life Span

Terminal PSI level of 2.5 is the average of the range of commonly accepted PSI levels for pavement maintenance used in the AASHTO design guide. The time of PSI declining from initial value to 2.5, namely pavement service life span of all highway segments before next maintenance is calculated under the hypothesis that AADT will not change, shown in Figure 6.

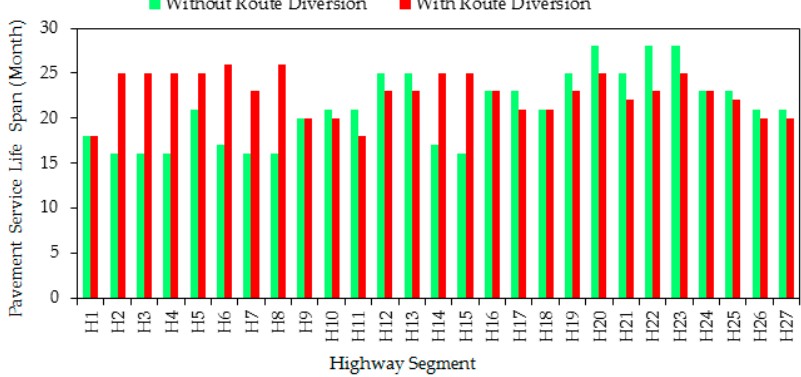

**Figure 6.** Pavement service life span.

From the results, route diversion can extend the service life span by an average of 5.4 months for G5, G93 and G76, but does not significantly shorten pavement service life span for other highways. The average pavement service life span of the total network is 22.8 months with route diversion, while 21.1 months without route diversion.

### 3.2.4. Average Travel Time

Because of route diversion, vehicles of class 5 to 11 do not drive along their shortest path, which increase their travel time. Figure 7 shows the comparison between average travel time with and without route diversion, which indicates that the average travel time of vehicles of class5 to 11 increases slightly by 6.2%.

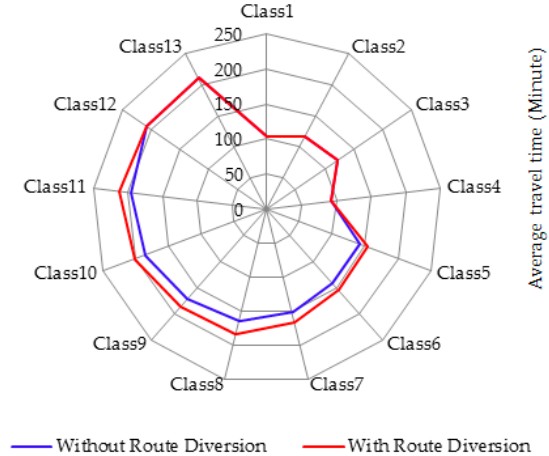

**Figure 7.** Average travel time.

### 3.2.5. Fuel Consumption and Pollutant Emissions

Fuel consumption of vehicles is influenced significantly by speed and pavement performance. Higher PSI can reduce fuel consumption and pollutant emissions. Figure 8 shows the comparison between average fuel consumption of 13 vehicles types with and without route diversion.

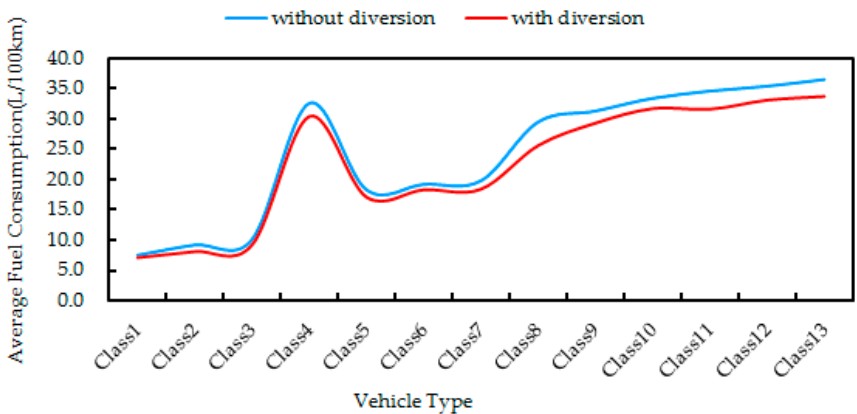

**Figure 8.** Average fuel consumption.

From Figure 8, we know that route diversion can save an average of 7.6% on fuel consumption for the 13 types of vehicles.

As for pollutant emissions, we take three main types of pollutants like $NO_X$, $CO_2$ and CO into consideration. From the results indicated by Figure 9, pollutant emissions can be reduced by an average of 6.8% on $NO_X$, 7.7% on $CO_2$ and 7.1% on CO with route diversion.

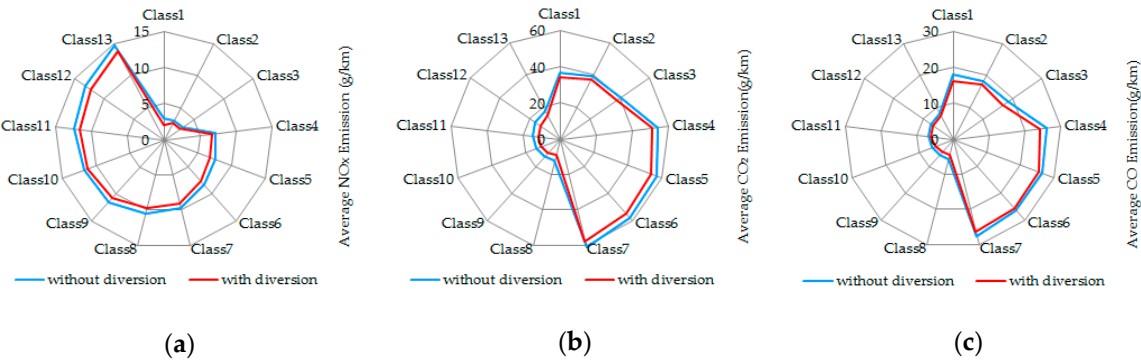

**Figure 9.** Average pollutant emission. (**a**) Average $NO_X$ emission; (**b**) Average $CO_2$ emission; (**c**) Average CO emission.

## 4. Discussion

$\theta$ is a weight factor for the first objective function $T$, $\theta \in [0,1]$, which decides the importance of objective functions in the execution of the model. The weight allocation has an influence on the calculation results of the model. Influence on average PSI and travel time of different valves of $\theta$ is discussed in the research. Figure 10 shows the variation of average PSI and travel time when $\theta$ belongs to $[0,1]$ with an interval of 0.1. From Figure 10, average travel time declines with the increase of $\theta$, while the variation of average PSI is an inverted U-shaped curve.

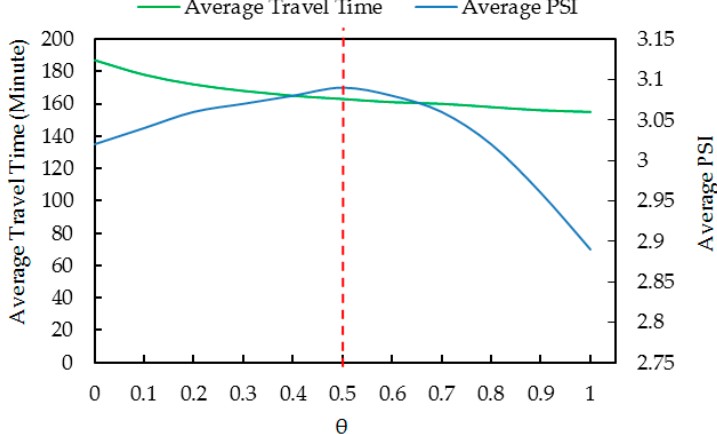

**Figure 10.** Sensitivity analysis of $\theta$.

When $\theta \leq 0.5$, objective $P$ is more considered. Smaller value of $\theta$ obtains more route diversion, which means more vehicles drive along a longer route with more travel time. From the inverted U-shaped curve, more route diversion does not mean higher average PSI, because lots of route diversion of vehicles to the adjoining highway may decline PSI of these highways significantly.

When $\theta \geq 0.5$, objective $T$ is more considered. Average travel time declines continuously, but its declining rate is lower than that when $\theta \leq 0.5$. Average PSI declines with a higher rate, because more vehicles drive along the current highways without route diversion causing a rapid decline of PSI of these highways.

From the sensitivity analysis, the value of $\theta$ has a great influence on the output of the model. It does mean that the bigger or the smaller $\theta$ is, the better the results are. It can be observed that when $\theta = 0.5$, average PSI peaks and declining rate of travel time starts to slow down. So, the optimal results can be obtained by the model when $\theta = 0.5$, which is why $\theta$ is set as 0.5 in this research.

## 5. Conclusions

Traffic load, especially trucks is the most important factor causing highway damage. Unbalance of traffic flow usually exists on several highways towards the same direction, and pavement performance of these highways with heavy traffic deteriorates more quickly, which may cost more maintenance expenditures. Existing DTA models do not take the influence on traffic assignment of pavement performance deterioration into consideration.

In this research, a multi-objective DTA model is proposed to assign traffic flow in the network with objectives of minimum travel time and minimum average PSI decline and a TLBO is coded to solve the model. Then, a case study was provided to confirm the effectiveness of the proposed model. The result shows that:

1.  Vehicles of class 5 to 11 are the main factors of pavement damage. To slow down PSI declining rate, an average of about 17.67% vehicles diverts their routes from G5, G93 and G76 to the adjoining highways, and vehicles of class 5, 8, 9, 10 account for about 75% of all diverting vehicles on both directions.

2.  The optimal traffic assignment obtained by the proposed model can save the total maintenance expenditures by 14.71%, extend the average pavement service life span of G5, G93 and G76 by 5.4 months, save the fuel consumption by an average of 7.6% and reduce pollutant emissions by an average of 6.8% on NOX, 7.7% on CO2 and 7.1% on CO in spite of a little increase of 6.2% on travel time of class 5 to 11.

This research provides a method to improve the sustainability of pavement performance by a reasonable traffic flow assignment in a network, which also contributes to the literature on dynamic traffic management. However, there are limitations in this research, for example, the model focuses on the highways with flexible pavements only, but does not consider other types of pavements in a network, which have different characteristic of performance deterioration. These limitations should be taken into consideration in the future work.

**Author Contributions:** X.M. designed research goals and wrote the manuscript; J.W. designed research methods; C.Y. collected the data; W.Y. revised and edited the manuscript and J.G. analyzed the data.

**Funding:** This research was funded by Humanities and Social Science Research Program of Ministry of Education in China (Grant Number 16XJCZH002) and supported by the Fundamental Research Funds for the Central Universities (Grant Number 310823170657 and 300102238501) and Social Science Major Theoretical and Practical Problems Research Project in Shaanxi province of China (Grant Number 2018Z013).

**Conflicts of Interest:** The authors declare no conflicts of interest.

## Abbreviations

| | |
|---|---|
| $T$ | Total travel time |
| $\{O\}$ | Set of origin nodes |
| $\{D\}$ | Set of destination nodes |
| $\{A\}$ | Set of direct links |
| $p$ | Arbitrary elements of $\{O\}$ |
| $q$ | Arbitrary elements of $\{D\}$ |
| $a$ | Arbitrary elements of $\{A\}$ |
| $v_a$ | Traffic flow on link $a$ |
| $s_a(v)$ | Traffic delay function for link $a$ |
| $K_{pq}$ | Set of paths from $p$ to $q$ |
| $k$ | Arbitrary element of $K_{pq}$ |
| $h_{k,pq}$ | Trips per unit time on path $k$ from $p$ to $q$ |
| $g_{pq}$ | Traffic demand per unit time from $p$ to $q$ |
| $\delta_{ak,pq}$ | If link $a$ lies on path $k$ from $p$ to $q$, $\delta_{ak,pq} = 1$, otherwise, $\delta_{ak,pq} = 0$ |
| $P_0$ | Initial PSI |
| $P_c$ | Terminal PSI of 2.5 |

| $\alpha, \beta$ | Coefficients |
|---|---|
| $N$ | Number of highway segments |
| $n$ | Number of vehicle types |
| $m$ | Number of axle weight groups |
| $Q_{ij}$ | Traffic volume of vehicle class $i$, which belongs to axle weight group $j$ |
| $LEF_{ij}$ | Load equivalency factor of weight group $j$ in vehicle class $i$ |
| $Q_{sij}$ | Traffic flow of vehicle class $i$ in axle weight group $j$ on highway segment $s$ |
| $P$ | Average PSI decline of all highway segments |
| $P_s{}^0$ | Initial PSI of highway segment $s$ |
| $P_s{}^c$ | Terminal PSI of highway segment $s$ |
| $LEF_{sij}$ | Load equivalency factor of axle weight group $j$ in vehicle class $i$ on segment $s$ |
| $W$ | Set of OD pairs |
| $K_w$ | Set of paths of OD pair $w$ |
| $h_{kwij}$ | Traffic flow of vehicle class $i$ in axle weight group $j$ on path $k$ of OD pair $w$ |
| $q_w$ | Traffic need of OD pair $w$ |
| $\delta_{swkij}$ | If highway segment $s$ with traffic flow of vehicle class $i$ in axle weight group, $j$ lies on path, $k$ of OD pair, $w$, $\delta_{swkij}$ =1, otherwise, $\delta_{swkij} = 0$ |
| $\tau_s$ | Lower limit value of PSI of highway segments $s$ |
| $\alpha_s, \beta_s$ | Coefficients of highway segment $s$ |
| $\theta$ | Weight factor for objective function |
| $P_{min}$ | Minimum values of $P$ |
| $T_{min}$ | Minimum values of $T$ |
| $D$ | Number of subjects |
| $V$ | Population of learners |
| $x_\phi{}^\gamma$ | Learner $\gamma$ in subject $\phi$ |
| $F(x)_{\phi,best}$ | Best learner in subject $\phi$ |
| $X_{\phi,teacher}$ | Chief teacher in subject $\phi$ |
| $M_\phi$ | Mean value of learners in subject $\phi$ |
| $M\_new_\phi$ | New mean value of learners in subject $\phi$ |
| $TF_\phi$ | Teaching factor |
| $Difference_\phi$ | Difference between $M_\phi$ and $M\_new_\phi$ |
| $x_{\phi,old}{}^\gamma$ | Old knowledge of learner $\gamma$ in subject $\phi$ |
| $x_{\phi,new}{}^\gamma$ | New knowledge of learner $\gamma$ in subject $\phi$ |
| $\overline{SV}$ | Mean value of slop variance in the wheel paths |
| $\overline{RD}$ | Mean rut depth |
| $C$ | Cracking |
| $G$ | Patching |
| $L$ | Length of the highway segment |
| $\sigma$ | Number of lanes of the highway segment |
| MOT | Ministry of Transport |
| ESAL | Equivalent number of standard axle |
| PSI | Present serviceability index |
| DTA | Dynamic traffic assignment |
| OD | Origin-destination |
| MPM | Mathematical programming model |
| OCM | Optimal control model |
| VIM | Variational inequality model |
| BPR | Bureau of Public Roads |
| FHWA | Federal Highway Administration |
| GA | Genetic Algorithm |

AIA          Artificial immune algorithm
ACO          Ant colony optimization
PSO          Particle swarm optimization
GSA          Gravitational search algorithm
TLBO         Teaching-learning-based optimization
CMA          Chengdu Metropolitan Area
SHAB         Sichuan Highway Administration Bureau
AADT         Annual average daily traffic
ATRs         Automatic traffic recorders
AVCs         Automatic vehicle classifiers
WIM          Weigh-in-motion
VLPR         Vehicle license plate recognition
AASHTO       American Association of State Highway and Transportation Officials

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
