# Peer review of "A Dynamic Traffic Assignment Model for the Sustainability of Pavement Performance"

_sustainability, doi:10.3390/su11010170_

Round 1

Reviewer 1 Report

The primary objective of the paper is to obtain a more optimized traffic assignment for pavement damage reduction by establishing a multi-objective DTA model with the objectives of not only minimum travel time but a minimum decline of Present Serviceability Index (PSI) for pavements. The proposed research is innovative and contains original features.

The paper is well written and clear.

Nonetheless, in the paper, there are some aspects that need additional clarification, explanation, or revision before the paper would be ready for publication.

1-      Small refinements in the text to improve the comprehension and the English language should be made.

a.       Line 214-216: I propose to modify the phrase: “The network was described as a topology graph including 11 nodes and 27 arcs (H1 to H27 highway segments), shown in Figure 3, where codes E1 to E12 refer to the external origins and destinations of vehicles, and nodes 1 to 11 indicate the 8 intersections of highways in the network.”

b.      Figure 3: Please, add in the figure E5 and E7 codes.

c.       Line 241-242: I propose to modify the phrase: “The coefficients a and b in Equation (5) have different values for different pavements. Because there is only flexible pavement in this case study, a and b were calibrated by using natural logarithm as follows”.

2-      In addition to the Average travel time evaluation with and without route diversion, fuel consumption and pollutant emissions could also be assessed.

3-      The conclusion should be improved by adding some numeric result of the study.

I think that the paper could be published in MDPI Sustainability after including the proposed reviews.

Author Response

Point 1: The primary objective of the paper is to obtain a more optimized traffic assignment for pavement damage reduction by establishing a multi-objective DTA model with the objectives of not only minimum travel time but a minimum decline of Present Serviceability Index (PSI) for pavements. The proposed research is innovative and contains original features.

The paper is well written and clear.

Nonetheless, in the paper, there are some aspects that need additional clarification, explanation, or revision before the paper would be ready for publication.

Response 1: The authors greatly appreciate the reviewer’s encouragement. This memo documents our responses to all review comments. The appropriate changes have been made to the manuscript.

Point 2: Line 214-216: I propose to modify the phrase: “The network was described as a topology graph including 11 nodes and 27 arcs (H1 to H27 highway segments), shown in Figure 3, where codes E1 to E12 refer to the external origins and destinations of vehicles, and nodes 1 to 11 indicate the 8 intersections of highways in the network.”. 

Response 2: Please refer to lines 219-221 of the refined manuscript, which have been rewritten by the authors according to reviewer’s suggestion.

Point 3: Figure 3: Please, add in the figure E5 and E7 codes.

Response 3: Please refer to Figure 3. The authors have added codes E5 and E7 in the figure.

Point 4: Line 241-242: I propose to modify the phrase: “The coefficients α and β in Equation (5) have different values for different pavements. Because there is only flexible pavement in this case study, α and β were calibrated by using natural logarithm as follows”.

Response 4: Please refer to lines 246-249. The authors have rewritten the paragraph according to reviewer’s suggestion and made some improvements to the paragraph.

Point 5: In addition to the Average travel time evaluation with and without route diversion, fuel consumption and pollutant emissions could also be assessed.

Response 5: The fuel consumption and pollutant emissions with and without route diversion has been analysed. A new section “3.2.5. Fuel Consumption and Pollutant Emissions” is added to the revised manuscript. Please refer to lines 321-339. Accordingly, the analysis results of fuel consumption and pollutant emissions have been added into both “Abstract” and “Conclusions”, please refer to lines 21-22 and lines 376-378.

Point 6: The conclusion should be improved by adding some numeric result of the study.

Response 6: The authors have added numeric results including the percentage of vehicles diversion, total maintenance expenditures savings, pavement service life span extension, fuel consumption savings, pollutant emissions reduction and travel time increase into the conclusion. Please refer to lines 370-378.

Reviewer 2 Report

Interesting paper that addresses an actual problem regarding sustainability. Only a few comments to improve the research work.

In section 3.1.2 it is not clear how alpha and beta parameters were obtained from field data. Due to the importance of this calibration procedure, it is suggested to enter in more detail on the description of how these two parameters were obtained.

As explicitly mentioned in the paper, the research work is limited to the case of flexible pavements. However, it is not sufficiently clear how different flexible pavement types (of highways) can be considered in the overall model of cumulative ESALs and decrease in PSI.

Author Response

Point 1: Interesting paper that addresses an actual problem regarding sustainability. Only a few comments to improve the research work.

Response 1: The authors greatly appreciate the reviewer’s encouragement. The authors have made an appropriate revision to the manuscript.

Point 2: In section 3.1.2 it is not clear how alpha and beta parameters were obtained from field data. Due to the importance of this calibration procedure, it is suggested to enter in more detail on the description of how these two parameters were obtained. 

Response 2: The data needed to calibrate the coefficients α and β is explained and how to collect the data is added to the revised manuscript. Please refer to lines 252-258.

Point 3: As explicitly mentioned in the paper, the research work is limited to the case of flexible pavements. However, it is not sufficiently clear how different flexible pavement types (of highways) can be considered in the overall model of cumulative ESALs and decrease in PSI.

Response 3: Equation (5) in the can be manuscript used to identify the correlation between PSI and cumulative ESALs. Coefficients α and β differ between different pavement types. The authors have revised Equation (8) in the model, which considers different pavement types using different α and β. Please refer to Equation (8).

Reviewer 3 Report

The authors' work aims to establish a multi-objective dynamic traffic assignment (DTA) that accomplishes two objectives: minimum travel time, but also minimum decline of Present Serviceability Index (PSI) for freeway pavements.

The manuscript is correctly written and structured, and the topic is quite interesting according to sustainability transportation perspective. However, this reviewer has to highlight some minor issues to improve the paper and it could finally be published:

- Think of adding a Nomenclature section/table.

- Grammar and spelling needs some polish, please revise it.

Some additional comments:

L34-36: A reference for these data is needed. Should be interesting to know the maintenance cost per km

L50: Define PSI acronym also in the main text.

L95: Change to "as in [33]"

Author Response

Point 1: The manuscript is correctly written and structured, and the topic is quite interesting according to sustainability transportation perspective. However, this reviewer has to highlight some minor issues to improve the paper and it could finally be published

Response 1: The authors greatly appreciate the reviewer’s encouragement. The authors have made an appropriate revision to the manuscript.

Point 2: Think of adding a Nomenclature section/table. 

Response 2: A new section “Appendix A” is added to the revised manuscript, which lists the symbols and abbreviations used in the manuscript. Please refer to lines 392-473.

Point 3: Grammar and spelling needs some polish, please revise it.

Response 3: The following revisions of grammar and spelling haven been done by authors.

Line 39: change “rate of pavement deterioration” to “pavement deterioration rate”

Line 45: change “a balance on the highways…” to “a balance between the highways…”

Line 79: change “more practical and easy” to “more practical and easier”

Line 108: change “in the same direction” to “towards the same direction”

Line 109: Change “single OD” to “a single OD”

Line 110: change “traffic demand from O to D is 2,000 vehicles” to “traffic demands from O to D are 2,000 vehicles”

Line 113: change “…, an unbalanced traffic distribution” to “causing an unbalanced traffic distribution”

Line 125: change “Equation (5) represents the correlation between PSI and cumulative ESALs” to “For different types of pavements, the correlation between PSI and cumulative ESALs can be identified using Equation (5)”

Line 129: change “as the follows” to “as follows”

Line 162: change “The proposed multi-objective DTA” to “The proposed multi-objective DTA model”

Line 208: change “as flexible pavements” to “as flexible pavements using asphalt mixture”

Line 217: change “a big financial burden” to “a heavy financial burden”

Line 228: change “pavement performance are needed” to “pavement performance is needed”

Line 246: change “the mean of slop” to “the mean value of slop”

Line 289: change “vehicles of class 5, 8, 9, 10 accounts for about 75%” to “vehicles of class 5, 8, 9, 10 account for about 75%”

Point 4: L34-36: A reference for these data is needed. Should be interesting to know the maintenance cost per km

Response 4: The authors have added a new reference to the manuscript, and please refer to reference 3. The average maintenance cost per km has been added, and please refer to line 36.

Point 5: L50: Define PSI acronym also in the main text.

Response 5: PSI acronym has been defined in the main text. Please refer to line 52.

Point 6: L95: Change to "as in [33]"

Response 6: Line 98 has been revised as “A basic DTA model is formulated as in[34]”.